# Derivation of the Langevin Equation from the Microcanonical Ensemble

**DOI:** 10.3390/e26040277

**Published:** 2024-03-22

**Authors:** Ralf Eichhorn

**Affiliations:** Nordita, Royal Institute of Technology and Stockholm University, 106 91 Stockholm, Sweden; eichhorn@nordita.org

**Keywords:** Langevin equation, Brownian motion, driven diffusion, microcanonical ensemble, fluctuation–dissipation relation

## Abstract

When writing down a Langevin equation for the time evolution of a “system” in contact with a thermal bath, one typically makes the implicit (and often tacit) assumption that the thermal environment is in equilibrium at all times. Here, we take this assumption as a starting point to formulate the problem of a system evolving in contact with a thermal bath from the perspective of the bath, which, since it is in equilibrium, can be described by the microcanonical ensemble. We show that the microcanonical ensemble of the bath, together with the Hamiltonian equations of motion for all the constituents of the bath and system together, give rise to a Langevin equation for the system evolution alone. The friction coefficient turns out to be given in terms of auto-correlation functions of the interaction forces between the bath particles and the system, and the Einstein relation is recovered. Moreover, the connection to the Fokker–Planck equation is established.

## 1. Introduction

The Langevin equation [1,2,3,4,5,6] is a well-established and extremely successful model for describing how a system evolves under the combined influence of deterministic and rapidly fluctuating (“random”) forces. Typically, the random forces and their stochastic nature are the result of many (microscopic) degrees of freedom, which change extremely quickly compared to the “slow” degrees of freedom of the system. A paradigmatic application is Brownian motion [2], i.e., the diffusive motion of a small particle in a fluid or gas, as schematically illustrated in Figure 1 (the Langevin equation for this setup is explicitly stated in (55)). Generally, a macroscopic thermodynamic system plays the role of a “thermal bath” or “environment” for a small system of interest. A common standard assumption when investigating the effective stochastic dynamics of such systems is that *the thermal bath remains in equilibrium at all times* [6].

This equilibrium assumption is justified by an extreme separation of time scales. The bath possesses a fast intrinsic time scale τ on which perturbations away from equilibrium decay rapidly (typically, τ is related to the molecular collision time, which is about $10^{-13}\;s$ in water). The dynamics of the system occur on time scales much slower than this characteristic time scale of the bath. For instance, for a colloidal particle, the fastest time scale is the velocity relaxation time due to viscous friction, which is several orders of magnitude larger than τ (it is about $10^{-8}\;s$ for a colloidal sphere of radius 0.1μm in water). Its (diffusive) spatial motion occurs on even slower time scales (about ms). Moreover, in typical experiments, external time-dependent variations in forces, etc., exerted on the system do not directly interfere with the bath and vary at times much larger than τ. Note that we completely disregard that perturbations might create slow collective “degrees of freedom” in the bath. Even though this is known to be the case in dense environments, the modifications of the system dynamics due to such slow collective modes in the bath seem to be of little practical relevance for typical systems usually modeled by Langevin equations. Put differently, the theory that we are going to develop here is a proper description for systems for which such slow bath modes do not have observable consequences.

Hence, at any instance of slow “system time”, i.e., at any time resolution Δt of interest in typical experiments that capture the system behavior (with Δt≫τ being microscopically large but macroscopically small), the bath is in an equilibrium state with a well-defined energy. Energy changes occur only via processes connected to changes in the system state, e.g., via a displacement of the colloidal particle. For the fast microscopic degrees of freedom of the bath. such processes, occurring over Δt time scales, are quasi-statically slow transformations from a given equilibrium state to a new one. Since the bath is isolated from the rest of the world, each of these equilibrium states can be characterized by the microcanonical ensemble for the current value of bath energy.

Based on a classical Hamiltonian description of the system and bath together (Section 2), and the representation of the bath state by the microcanonical ensemble, we analyze the *system* dynamics on Δt time scales. We show that individual Δt steps are statistically independent (Section 3.1), implying that the system dynamics are Markovian on Δt time scales. For individual time steps Δt, we calculate averages and correlations of changes in the system state (Section 3.2). In linear order Δt, these moments turn out to be identical to those generated by a standard Langevin equation for the system dynamics alone (Section 3.3). Hence, we demonstrate that the standard Langevin equation represents an “effective” evolution equation for a system in contact with a bath, both together described on the microscopic level by a classical Hamiltonian. We also derive from the Hamiltonian description that the proper rules of stochastic calculus [3,4] for the effective Langevin equation are given by the Stratonovich interpretation (Section 3.4).

## 2. Model

We consider the setup schematically illustrated in Figure 1. A macroscopically large thermodynamic system plays the role of a thermal bath or environment for a (small) system of interest. We denote all parameters and coordinates, etc., related to the bath by small letters, and those specifying the system by capital letters; Greek indices denote particle numbers and Latin indices denote components of three-dimensional (coordinate) vectors. We collect all bath degrees of freedom in the super-vector $\phi =\left(p,q\right)=\left(p_{1},p_{2},\ldots ,p_{n},q_{1},q_{2},\ldots ,q_{n}\right)$ and all system degrees of freedom in Φ=(P,Q)$=(P_{1},P_{2},\ldots ,P_{N},Q_{1},Q_{2},\ldots ,Q_{N})$. We assume that N⋘n. The total Hamiltonian of the bath and system together is
(1a)$$H\left(\phi ,\Phi \right)=\underset{=H_{\mathrm{bath}}\left(\phi ,Q\right)}{\underset{︸}{H_{\mathrm{bulk}}\left(\phi \right)+H_{\mathrm{int}}\left(q,Q\right)}}+H_{\mathrm{sys}}\left(\Phi ,\lambda \right)\;,$$
with
(1b)$$\begin{array}{cc} H_{\mathrm{bulk}}\left(\phi \right) & =\sum_{\mu }\frac{p_{\mu }^{2}}{2m}+\sum_{\mu <\nu }U_{\mathrm{bulk}}\left(\right|q_{\mu }-q_{\nu }\left|\right)\;, \end{array}$$
(1c)$$\begin{array}{cc} H_{\mathrm{int}}\left(q,Q\right) & =\sum_{\mu ,\nu }U_{\mathrm{int}}\left(\right|q_{\mu }-Q_{\nu }\left|\right)\;, \end{array}$$
(1d)$$\begin{array}{cc} H_{\mathrm{sys}}\left(\Phi ,\lambda \right) & =\sum_{\mu }\frac{P_{\mu }^{2}}{2M}+U\left(Q,\lambda \right)\;. \end{array}$$
The interactions between the bath particles in the bulk are captured in $U_{\mathrm{bulk}}\left(\right|q_{\mu }-q_{\nu }\left|\right)$, and the interaction between bath particles and the system is captured in $U_{\mathrm{int}}\left(\right|q_{\mu }-Q_{\nu }\left|\right)$. Both depend on particle–particle distances. We combine $H_{\mathrm{bulk}}\left(\phi \right)$ and $H_{\mathrm{int}}\left(\phi ,Q\right)$ into the bath Hamiltonian $H_{\mathrm{bath}}\left(\phi ,Q\right)$ because, for the effective system dynamics that we are aiming at, the system energy should be determined by $H_{\mathrm{sys}}\left(\Phi ,\lambda \right)$ alone without reference to bath degrees of freedom. The potential U(Q,λ) represents the potential forces applied to the system by external means; they affect only positional degrees of freedom (not momenta). The potential is, moreover, allowed to depend on time via the protocol λ=λ(t), varying noticeably only on time scales much larger than τ.

### 2.1. Dynamics

The Hamilton equations of motion are [7]
(2a)$$\begin{array}{cc} {\dot{p}}_{\mu } & =-\frac{\partial H}{\partial q_{\mu }}=-\frac{\partial H_{\mathrm{bath}}}{\partial q_{\mu }}\;, \end{array}$$
(2b)$$\begin{array}{cc} {\dot{q}}_{\mu } & =\frac{\partial H}{\partial p_{\mu }}=\frac{p_{\mu }}{m}\;, \end{array}$$
(2c)$$\begin{array}{cc} {\dot{P}}_{\mu } & =-\frac{\partial H}{\partial Q_{\mu }}=-\frac{\partial H_{\mathrm{int}}}{\partial Q_{\mu }}-\frac{\partial H_{\mathrm{sys}}}{\partial Q_{\mu }}\;, \end{array}$$
(2d)$$\begin{array}{cc} {\dot{Q}}_{\mu } & =\frac{\partial H}{\partial P_{\mu }}=\frac{P_{\mu }}{M}\;, \end{array}$$
where the partial derivative symbols with respect to a vector quantity denote the corresponding gradient vector. Following standard convention in physics, we use the same symbols for the solutions of these equations as for the coordinates in phase space (ϕ,Φ), just endowing them with a time argument, which we write here as an index; if useful, we will also explicitly state the dependence on the initial values. The formal solutions for the various components $p=(p_{1},p_{2},\ldots ,p_{n})$, $q=(q_{1},q_{2},\ldots ,q_{n})$ and $P=(P_{1},P_{2},\ldots ,P_{N})$, $Q=(Q_{1},Q_{2},\ldots ,Q_{N})$ are
(3a)$$\begin{array}{cc} p_{t}\left(\phi_{0},\Phi_{0}\right) & =p_{0}-\int_{t_{0}}^{t}dt^{\prime }\;\frac{\partial H_{\mathrm{bath}}\left(\phi_{t^{\prime }},Q_{t^{\prime }}\right)}{\partial q}\;, \end{array}$$
(3b)$$\begin{array}{cc} q_{t}\left(\phi_{0},\Phi_{0}\right) & =q_{0}+\int_{t_{0}}^{t}dt^{\prime }\;\frac{p_{t^{\prime }}}{m}\;, \end{array}$$
(3c)$$\begin{array}{cc} P_{t}\left(\phi_{0},\Phi_{0}\right) & =P_{0}-\int_{t_{0}}^{t}dt^{\prime }\;\left[\frac{\partial H_{\mathrm{int}}\left(q_{t^{\prime }},Q_{t^{\prime }}\right)}{\partial Q}+\frac{\partial H_{\mathrm{sys}}\left(\Phi_{t^{\prime }},\lambda_{t^{\prime }}\right)}{\partial Q}\right]\;, \end{array}$$
(3d)$$\begin{array}{cc} Q_{t}\left(\phi_{0},\Phi_{0}\right) & =Q_{0}+\int_{t_{0}}^{t}dt^{\prime }\;\frac{P_{t^{\prime }}}{M}\;, \end{array}$$
for an initial configuration $\left(\phi_{0},\Phi_{0}\right)$ at time $t_{0}$. Here, the notation $\frac{\partial H_{\mathrm{bath}}\left(\phi_{t^{\prime }},Q_{t^{\prime }}\right)}{\partial q}$ means that we evaluate the derivative of $H_{\mathrm{bath}}\left(\phi ,Q\right)$ with respect to *q* (this is the 3n-dimensional gradient vector) and evaluate it along the trajectory $\left(\phi_{t^{\prime }},Q_{t^{\prime }}\right)$, and likewise for similar expressions.

According to the time scale separation discussed above, $\left(q_{t},p_{t}\right)$ vary extremely rapidly on the molecular collision time scale τ, while $\left(Q_{t},P_{t}\right)$ are virtually constant on that scale but change quickly on macroscopic time scales. The system degrees of freedom $\left(Q_{t},P_{t}\right)$ therefore evolve on a mesoscopic time scale Δt with τ≪Δt≪(macroscopic time scale). For the time evolution over such a time interval Δt, we can write
(4)$$\begin{array}{cc} Q_{\Delta t}\left(\phi ,\Phi \right) & =Q+\int_{t}^{t+\Delta t}dt^{\prime }\;\frac{P_{t^{\prime }}}{M}\\ & =Q+\frac{P}{M}\Delta t+O\left(\Delta t^{2}\right)=Q+O\left(\Delta t\right)\;, \end{array}$$
with *Q* denoting the system configuration at time *t*. A similar expansion in powers of Δt is not possible for $q_{t}$ (or $p_{t}$) due to its rapid fluctuations. Likewise, $P_{t}$ cannot be directly expanded in Δt because it is governed by interactions with the fast bath degrees of freedom (see Equation (3c)).

### 2.2. Statistical Mechanics

The total energy of the bath and system together,
(5)$$E_{\mathrm{tot}}\left(\lambda \right)=H_{\mathrm{bulk}}\left(\phi \right)+H_{\mathrm{int}}\left(q,Q\right)+H_{\mathrm{sys}}\left(\Phi ,\lambda \right)\;,$$
is conserved by the dynamics (2) for fixed λ. In the general case of external time-dependent protocols λ=λ(t), the energy is not conserved. However, only slow changes in λ(t) over mesoscopic time intervals Δt≫τ are permitted (and relevant experimentally). From the perspective of the bath, variations in λ are thus quasi-statically slow, such that the bath has a well-defined equilibrium energy within each time interval Δt. This energy is determined by the current values of Φ,
(6)$$\begin{array}{cc} E=E(\Phi ) & =E_{\mathrm{tot}}\left(\lambda \right)-H_{\mathrm{sys}}\left(\Phi ,\lambda \right)\\ & =H_{\mathrm{bulk}}\left(\phi \right)+H_{\mathrm{int}}\left(q,Q\right)\\ & =H_{\mathrm{bath}}\left(\phi ,Q\right)\;. \end{array}$$
Note that if all parameters (ϕ,Φ) remain fixed and only λ is varied, the total energy is changed via the contribution in the system potential, while the bath energy remains unchanged (and therefore does not directly depend on λ).

According to our equilibrium assumption, the bath can be described by the microcanonical ensemble, which is consistent with the current values of Φ and λ. The corresponding microcanonical partition function Ω(E) is given by [8]
(7)$$\Omega \left(E\right)=\Omega \left(E\left(\Phi \right)\right)=\int d\phi \;\delta \left(E\left(\Phi \right)-H_{\mathrm{bath}}\left(\phi ,Q\right)\right)\;.$$
According to standard statistical mechanics [8], the microcanonical density is
(8)$$\rho_{\mathrm{mc}}\left(\phi \right)=\frac{\delta \left(E-H_{\mathrm{bath}}\left(\phi ,Q\right)\right)}{\Omega (E)}\;,$$
and the temperature *T* of the bath is defined as ($k_{B}$ is Boltzmann’s constant)
(9)$$\frac{1}{k_{B}T}=\frac{\partial }{\partial E}\ln\Omega \left(E\right)=\frac{\Omega^{\prime }\left(E\right)}{\Omega (E)}\;.$$

## 3. Effective System Dynamics

Starting from an initial state of the bath and system together, the specific evolution of the system state Φ according to (2c), (2d) is affected by the rapidly fluctuating bath degrees of freedom, uniquely emerging from their initial state (and the one of the system). Different initial states of the bath thus lead to different “realizations” of the system dynamics, even if they start from the same initial configuration. Since we do not know the initial state of the bath exactly, but only that it is consistent with the microcanonical density (Equation 8), we can write the transition probability density for the system to evolve from Φ=(P,Q) to $\Phi^{\prime }=\left(P^{\prime },Q^{\prime }\right)$ during a mesoscopic time interval Δt as
(10)$$p_{E}\left(\Phi^{\prime }|\Phi \right)=\frac{\int d\phi \;\delta \left(E-H_{\mathrm{bath}}\left(\phi ,Q\right)\right)\;\delta \left(\Phi^{\prime }-\Phi_{\Delta t}\left(\phi ,\Phi \right)\right)}{\Omega (E)}\;,$$
where we denote the initial state of the bath at the beginning of the time interval by the unprimed variable ϕ. This probability density gives rise to the statistical properties of “small displacements” $\Delta \Phi =\Phi^{\prime }-\Phi$ of the system degrees of freedom. Their analysis is the central aim of this paper. Before evaluating averages and (co-)variances in Section 3.2, we start in Section 3.1 with demonstrating that the system displacements within successive time steps Δt are independent.

### 3.1. Markov Property

We consider a configuration (ϕ,Φ) of the bath and system together, which, from time *t*, evolves for a duration 2Δt according to the full microscopic dynamics (2). Generalizing (Equation 10), the probability density for finding the system in configuration $\Phi^{\prime }$ at time t+Δt and in configuration $\Phi^{\prime \prime }$ at time t+2Δt when it started at Φ at time *t* is
(11)$$\begin{array}{cc} & p(\Phi^{\prime \prime },t+2\Delta t;\Phi^{\prime },t+\Delta t|\Phi ,t)\\ & \;=\frac{1}{\Omega (E)}\int d\phi \;\delta \left(E-H_{\mathrm{bath}}\left(\phi ,Q\right)\right)\;\delta \left(\Phi^{\prime }-\Phi_{\Delta t}\left(\phi ,\Phi \right)\right)\;\delta \left(\Phi^{\prime \prime }-\Phi_{2\Delta t}\left(\phi ,\Phi \right)\right)\;. \end{array}$$
We will show in the following that $p(\Phi^{\prime \prime },t+2\Delta t;\Phi^{\prime },t+\Delta t|\Phi ,t)$ fulfills the Markov property
(12)$$p\left(\Phi^{\prime \prime },t+2\Delta t;\Phi^{\prime },t+\Delta t|\Phi ,t\right)=p\left(\Phi^{\prime \prime },t+2\Delta t|\Phi^{\prime },t+\Delta t\right)p\left(\Phi^{\prime },t+\Delta t|\Phi ,t\right)\;,$$
where
(13a)$$\begin{array}{cc} p(\Phi^{\prime },t+\Delta t|\Phi ,t) & =\frac{\int d\phi \;\delta \left(E-H_{\mathrm{bath}}\left(\phi ,Q\right)\right)\;\delta \left(\Phi^{\prime }-\Phi_{\Delta t}\left(\phi ,\Phi \right)\right)}{\Omega (E)}\;, \end{array}$$
(13b)$$\begin{array}{cc} p(\Phi^{\prime \prime },t+2\Delta t|\Phi^{\prime },t+\Delta t) & =\frac{\int d\phi^{\prime }\;\delta \left(E^{\prime }-H_{\mathrm{bath}}\left(\phi^{\prime },Q^{\prime }\right)\right)\;\delta \left(\Phi^{\prime \prime }-\Phi_{\Delta t}\left(\phi^{\prime },\Phi^{\prime }\right)\right)}{\Omega (E^{\prime })}\;, \end{array}$$
in analogy to (Equation 10) (we skip the energy subscript at *p* for notational convenience). The energy $E^{\prime }$ in (13b) is given by the energy balance (Equation 6) for the primed variables, and with λ=λ(t+Δt).

Starting from the expression (Equation 11), we first use the composition property of solutions to the Hamiltonian equations of motion to write $\delta \left(\Phi^{\prime \prime }-\Phi_{2\Delta t}\left(\phi ,\Phi \right)\right)=\delta \left(\Phi^{\prime \prime }-\Phi_{\Delta t}\left(\phi_{\Delta t}\left(\phi ,\Phi \right),\Phi_{\Delta t}\left(\phi ,\Phi \right)\right)\right)$. Then, we insert unity into the integral, expressed in the form $\int d\phi^{\prime }\;\delta \left(\phi^{\prime }-\phi_{\Delta t}\left(\phi ,\Phi \right)\right)$. With the presence of this delta function in the integrand, we can replace $\phi_{\Delta t}\left(\phi ,\Phi \right)$ everywhere by $\phi^{\prime }$, and likewise for $\Phi^{\prime }$ due to the delta function $\delta \left(\Phi^{\prime }-\Phi_{\Delta t}\left(\phi ,\Phi \right)\right)$. These steps turn (Equation 11) into
(14)$$\begin{array}{cc} & p(\Phi^{\prime \prime },t+2\Delta t;\Phi^{\prime },t+\Delta t|\Phi ,t)\\ & =\frac{1}{\Omega (E)}\int d\phi \;\delta \left(E-H_{\mathrm{bath}}\left(\phi ,Q\right)\right)\;\delta \left(\Phi^{\prime }-\Phi_{\Delta t}\left(\phi ,\Phi \right)\right)\\ & \;\;\;\;\times \int d\phi^{\prime }\;\delta \left(\phi^{\prime }-\phi_{\Delta t}\left(\phi ,\Phi \right)\right)\;\delta \left(\Phi^{\prime \prime }-\Phi_{\Delta t}\left(\phi^{\prime },\Phi^{\prime }\right)\right)\\ & =\int d\phi^{\prime }\;\delta \left(\Phi^{\prime \prime }-\Phi_{\Delta t}\left(\phi^{\prime },\Phi^{\prime }\right)\right)\\ & \;\;\;\times \frac{1}{\Omega (E)}\int d\phi \;\delta \left(E-H_{\mathrm{bath}}\left(\phi ,Q\right)\right)\;\delta \left(\phi^{\prime }-\phi_{\Delta t}\left(\phi ,\Phi \right)\right)\;\delta \left(\Phi^{\prime }-\Phi_{\Delta t}\left(\phi ,\Phi \right)\right)\;, \end{array}$$
exchanging the order of integration in the second equality. Here, the second line is the probability density $p(\phi^{\prime },\Phi^{\prime },t+\Delta t|\Phi ,t)$ for the total system to be in the state $\left(\phi^{\prime },\Phi^{\prime }\right)$ after evolving for time Δt, given that the system is in state Φ at time *t*. Hence,
(15)$$p\left(\Phi^{\prime \prime },t+2\Delta t;\Phi^{\prime },t+\Delta t|\Phi ,t\right)=\int d\phi^{\prime }\;\delta \left(\Phi^{\prime \prime }-\Phi_{\Delta t}\left(\phi^{\prime },\Phi^{\prime }\right)\right)\;p\left(\phi^{\prime },\Phi^{\prime },t+\Delta t|\Phi ,t\right)\;.$$

So far, the calculations are exact. Now, however, we have to make use of our central assumption that the bath is in equilibrium at any time point *t*, t+Δt, t+2Δt, etc., with a bath energy that is consistent with the current state of the system. This implies that the probability density of bath configurations $\phi^{\prime }$ depends only on the system state at the same time point, but not on its state at earlier times (this is used in the second step below), and that it is given by the microcanonical density (used in the third step, cf. Equation (Equation 8)). We can thus write
(16)$$\begin{array}{cc} p(\phi^{\prime },\Phi^{\prime },t+\Delta t|\Phi ,t) & =p\left(\phi^{\prime },t+\Delta t|\Phi^{\prime },t+\Delta t;\Phi ,t\right)\;p\left(\Phi^{\prime },t+\Delta t|\Phi ,t\right)\\ & =p\left(\phi^{\prime },t+\Delta t|\Phi^{\prime },t+\Delta t\right)\;p\left(\Phi^{\prime },t+\Delta t|\Phi ,t\right)\\ & =\frac{\delta \left(E^{\prime }-H_{\mathrm{bath}}\left(\phi^{\prime },Q^{\prime }\right)\right)}{\Omega (E^{\prime })}\;p\left(\Phi^{\prime },t+\Delta t|\Phi ,t\right)\;. \end{array}$$
Plugging this relation into (Equation 15), we obtain
(17)$$\begin{array}{cc} & p(\Phi^{\prime \prime },t+2\Delta t;\Phi^{\prime },t+\Delta t|\Phi ,t)\\ & \;=p\left(\Phi^{\prime },t+\Delta t|\Phi ,t\right)\;\frac{1}{\Omega (E^{\prime })}\int d\phi^{\prime }\;\delta \left(E^{\prime }-H_{\mathrm{bath}}\left(\phi^{\prime },Q^{\prime }\right)\right)\;\delta \left(\Phi^{\prime \prime }-\Phi_{\Delta t}\left(\phi^{\prime },\Phi^{\prime }\right)\right)\\ & \;=p\left(\Phi^{\prime },t+\Delta t|\Phi ,t\right)\;p\left(\Phi^{\prime \prime },t+2\Delta t|\Phi^{\prime },t+\Delta t\right)\;. \end{array}$$
In the second equality, we used the relation (13b), completing the proof of the Markov property (Equation 12) for the effective system dynamics on Δt time scales.

### 3.2. Average System Behavior

The main idea is to calculate from (Equation 10) the average displacements $\langle \Phi^{\prime }-\Phi \rangle =\langle \Delta \Phi \rangle$ and various (co-)variances 〈ΔΦΔΦ〉, in linear order in Δt, and to assess which kind of stochastic evolution equation for the system dynamics alone would produce the same moments in leading order Δt. According to our result from the previous section, it is sufficient to consider any such time interval Δt because the system dynamics are Markovian.

For clarity, we consider just one system particle such that Φ=(P,Q)=(P,Q) with $P=P=(P_{1},P_{2},P_{3})$ and $Q=Q=(Q_{1},Q_{2},Q_{3})$. We then have to evaluate (i,j=1,2,3)
(18a)$$\begin{array}{cc} \langle \Delta Q_{i}\rangle & =\langle Q_{i}^{\prime }-Q_{i}\rangle \\ & =\int d\Phi^{\prime }\;p_{E}\left(\Phi^{\prime }|\Phi \right)\left(Q_{i}^{\prime }-Q_{i}\right)\;,\\ \langle \Delta P_{i}\rangle & =\langle P_{i}^{\prime }-P_{i}\rangle \end{array}$$
(18b)$$\begin{array}{cc} & =\int d\Phi^{\prime }\;p_{E}\left(\Phi^{\prime }|\Phi \right)\left(P_{i}^{\prime }-P_{i}\right)\;, \end{array}$$
(18c)$$\begin{array}{cc} \langle \Delta Q_{i}\Delta Q_{j}\rangle & =\int d\Phi^{\prime }\;p_{E}\left(\Phi^{\prime }|\Phi \right)\left(Q_{i}^{\prime }-Q_{i}\right)\left(Q_{j}^{\prime }-Q_{j}\right)\;, \end{array}$$
(18d)$$\begin{array}{cc} \langle \Delta P_{i}\Delta P_{j}\rangle & =\int d\Phi^{\prime }\;p_{E}\left(\Phi^{\prime }|\Phi \right)\left(P_{i}^{\prime }-P_{i}\right)\left(P_{j}^{\prime }-P_{j}\right)\;, \end{array}$$
(18e)$$\begin{array}{cc} \langle \Delta Q_{i}\Delta P_{j}\rangle & =\int d\Phi^{\prime }\;p_{E}\left(\Phi^{\prime }|\Phi \right)\left(Q_{i}^{\prime }-Q_{i}\right)\left(P_{j}^{\prime }-P_{j}\right)\;, \end{array}$$
to lowest order in Δt. Note that the averages in (18) are over the final configuration $\Phi^{\prime }$ only, conditioned on a fixed (but arbitrary) initial value Φ.

#### 3.2.1. Evaluation of $\langle \Delta Q_{i}\rangle$

We use the expression (Equation 10) for $p_{E}\left(\Phi^{\prime }|\Phi \right)$ in ([Disp-formula FD18a-entropy-26-00277]):(19)$$\begin{array}{cc} \langle \Delta Q_{i}\rangle & =\int d\Phi^{\prime }\;\left(Q_{i}^{\prime }-Q_{i}\right)\;\frac{\int d\phi \;\delta \left(E-H_{\mathrm{bath}}\left(\phi ,Q\right)\right)\;\delta \left(\Phi^{\prime }-\Phi_{\Delta t}\left(\phi ,\Phi \right)\right)}{\Omega (E)}\\ & =\frac{1}{\Omega (E)}\int d\phi \;\left[{\left(Q_{i}\right)}_{\Delta t}\left(\phi ,\Phi \right)-Q_{i}\right]\;\delta \left(E-H_{\mathrm{bath}}\left(\phi ,Q\right)\right)\;. \end{array}$$
Since ${\left(Q_{i}\right)}_{\Delta t}\left(\phi ,\Phi \right)$ is a slowly varying function of *t* on Δt time intervals, we can use the expansion from (Equation 4) and obtain
(20)$$\begin{array}{cc} \langle \Delta Q_{i}\rangle & =\frac{1}{\Omega (E)}\int d\phi \;\left[Q_{i}+\frac{P_{i}}{M}\Delta t-Q_{i}\right]\;\delta \left(E-H_{\mathrm{bath}}\left(\phi ,Q\right)\right)+O\left(\Delta t^{2}\right)\\ & =\frac{P_{i}}{M}\Delta t+O\left(\Delta t^{2}\right)\;. \end{array}$$
The linear order result is therefore, as expected,
(21)$$\langle \Delta Q_{i}\rangle =\frac{P_{i}}{M}\Delta t\;.$$

#### 3.2.2. Evaluation of $\langle \Delta P_{i}\rangle$

We start again by using the expression (Equation 10) for $p_{E}\left(\Phi^{\prime }|\Phi \right)$,
(22)$$\begin{array}{cc} \langle \Delta P_{i}\rangle & =\int d\Phi^{\prime }\;\left(P_{i}^{\prime }-P_{i}\right)\;\frac{\int d\phi \;\delta \left(E-H_{\mathrm{bath}}\left(\phi ,Q\right)\right)\;\delta \left(\Phi^{\prime }-\Phi_{\Delta t}\left(\phi ,\Phi \right)\right)}{\Omega (E)}\\ & =\frac{1}{\Omega (E)}\int d\phi \;\left[{\left(P_{i}\right)}_{\Delta t}\left(\phi ,\Phi \right)-P_{i}\right]\;\delta \left(E-H_{\mathrm{bath}}\left(\phi ,Q\right)\right)\\ & =-\frac{1}{\Omega (E)}\int d\phi \;\int_{t}^{t+\Delta t}dt^{\prime }\;\left[\frac{\partial H_{\mathrm{int}}\left(q_{t^{\prime }},Q_{t^{\prime }}\right)}{\partial Q_{i}}+\frac{\partial H_{\mathrm{sys}}\left(\Phi_{t^{\prime }},\lambda_{t^{\prime }}\right)}{\partial Q_{i}}\right]\;\delta \left(E-H_{\mathrm{bath}}\left(\phi ,Q\right)\right)\;, \end{array}$$
where, in the last line, we inserted the formal solution from (3c). The system contribution (second term in the brackets) is a slowly varying function of time on Δt scales. Defining the external force on the system,
(23)$$f\left(Q,\lambda \right)=-\frac{\partial U(Q,\lambda )}{\partial Q}=-\frac{\partial H_{\mathrm{sys}}\left(\Phi ,\lambda \right)}{\partial Q}\;,$$
and expanding $Q_{t^{\prime }}=Q+\left(P/M\right)\left(t^{\prime }-t\right)+\ldots$ as above and $\lambda_{t^{\prime }}$ as $\lambda_{t^{\prime }}=\lambda_{t^{\prime }=t}+{\dot{\lambda }}_{t^{\prime }=t}\left(t^{\prime }-t\right)+\ldots$, we find $-\frac{\partial H_{\mathrm{sys}}\left(\Phi_{t^{\prime }},\lambda_{t^{\prime }}\right)}{\partial Q_{i}}=f_{i}\left(Q,\lambda \right)+O\left(t^{\prime }-t\right)$ (with $\lambda =\lambda_{t^{\prime }=t}$). Using this expansion, we can perform the time integral and the microcanonical average for the system contribution. We obtain for (Equation 22)
(24)$$\langle \Delta P_{i}\rangle =f_{i}\left(Q,\lambda \right)\Delta t-\frac{1}{\Omega (E)}\int d\phi \;\int_{t}^{t+\Delta t}dt^{\prime }\;\frac{\partial H_{\mathrm{int}}\left(q_{t^{\prime }},Q_{t^{\prime }}\right)}{\partial Q_{i}}\;\delta \left(E-H_{\mathrm{bath}}\left(\phi ,Q\right)\right)+O\left(\Delta t^{2}\right)\;.$$

A similar expansion procedure is not achievable for the interaction part because $H_{\mathrm{int}}\left(q_{t^{\prime }},Q_{t^{\prime }}\right)=\sum_{\mu }U_{\mathrm{int}}\left(|{\left(q_{\mu }\right)}_{t^{\prime }}-Q_{t^{\prime }}|\right)$ is a highly fluctuating function of time. The contribution $Q_{t^{\prime }}\approx Q+\left(P/M\right)\left(t^{\prime }-t\right)$ just superimposes a slow drift over the rapid fluctuations. Nevertheless, in order to analyze this term, we can start by evaluating the “zeroth-order” effect by approximating $Q_{t^{\prime }}$ with its initial value, $Q_{t}=Q$. The interaction term is then
(25)$$\begin{array}{c} -\frac{1}{\Omega (E)}\int_{t}^{t+\Delta t}dt^{\prime }\;\int d\phi \;\frac{\partial H_{\mathrm{int}}\left(q_{t^{\prime }},Q_{t^{\prime }}\right)}{\partial Q_{i}}\;\delta \left(E-H_{\mathrm{bath}}\left(\phi ,Q\right)\right)\\ \;\approx -\frac{1}{\Omega (E)}\int_{t}^{t+\Delta t}dt^{\prime }\;\int d\phi \;\frac{\partial H_{\mathrm{int}}\left(q_{t^{\prime }},Q\right)}{\partial Q_{i}}\;\delta \left(E-H_{\mathrm{bath}}\left(\phi ,Q\right)\right)\;. \end{array}$$

The ϕ integral
(26)$$\frac{1}{\Omega (E)}\int d\phi \;\delta \left(E-H_{\mathrm{bath}}\left(\phi ,Q\right)\right)\;\frac{\partial H_{\mathrm{int}}\left(q_{t^{\prime }},Q\right)}{\partial Q_{i}}={\langle \frac{\partial H_{\mathrm{int}}\left(q_{t^{\prime }},Q\right)}{\partial Q_{i}}\rangle }_{\;\;\mathrm{mc}}$$
represents the momentary, average net force (component *i*) that the bath exerts on an immobile system particle. For physical reasons, we expect this force to vanish, as there should be no net force on the particle from the thermal environment. In Appendix A, we prove that this is indeed the case.

The question now arises as to whether there will be a net Δt contribution from
(27)$$-\frac{1}{\Omega (E)}\int_{t}^{t+\Delta t}dt^{\prime }\;\int d\phi \;\delta \left(E-H_{\mathrm{bath}}\left(\phi ,Q\right)\right)\;\frac{\partial H_{\mathrm{int}}\left(q_{t^{\prime }},Q_{t^{\prime }}\right)}{\partial Q_{i}}\;,$$
when considering the next order $Q_{t^{\prime }}\approx Q+\left(P/M\right)\left(t^{\prime }-t\right)$ in the particle displacement. In this case, the particle moves (slowly) relative to the bath and we indeed expect a frictional contribution to show up, which, in lowest order, should be proportional to (P/M)Δt (see also the illuminating discussion in Section 15.5 of [8]). Since we cannot expand $\frac{\partial H_{\mathrm{int}}\left(q_{t^{\prime }},Q\right)}{\partial Q_{i}}$ in $t^{\prime }$, we extract the leading contribution by a partial integration in the time integral. Moreover, from a physical perspective, it will be instructive to transform to coordinates co-moving with the system particle. We will perform the calculation with and without the transformation to the co-moving frame, which will result in two alternative (but equivalent) expressions for the prefactor of (P/M)Δt.

##### Co-Moving Frame

In order to switch to the co-moving frame, we introduce for all μ=1,2,…,n the time-dependent coordinate transformation (for fixed initial time *t* at which the bath and system are in the state $\left(\phi_{t},\Phi_{t}\right)=\left(\phi ,\Phi \right)$)
(28)$${\tilde{q}}_{\mu }\left(t^{\prime }\right)=q_{\mu }-\left(Q_{t^{\prime }}-Q\right)\approx q_{\mu }-\frac{P}{M}\left(t^{\prime }-t\right)\Leftrightarrow q_{\mu }={\tilde{q}}_{\mu }\left(t^{\prime }\right)+\frac{P}{M}\left(t^{\prime }-t\right)\;.$$
Differences between bath particle coordinates are invariant under this transformation, implying that
(29a)$$H_{\mathrm{bulk}}\left(p,q\right)=H_{\mathrm{bulk}}\left(p,\tilde{q}\left(t\right)\right)\;.$$
Differences between bath particle coordinates and the system coordinate transform according to
(29b)$$\begin{array}{cc} q_{\mu }-Q & ={\tilde{q}}_{\mu }\left(t^{\prime }\right)-\left(Q-\frac{P}{M}\left(t^{\prime }-t\right)\right)\;,\\ {\left(q_{\mu }\right)}_{t^{\prime }}-Q_{t^{\prime }} & =q_{\mu }+\int_{t}^{t^{\prime }}dt^{\prime \prime }\;\frac{{\left(p_{\mu }\right)}_{t^{\prime \prime }}}{m}-\left(Q+\frac{P}{M}\left(t^{\prime }-t\right)\right) \end{array}$$
(29c)$$\begin{array}{cc} & ={\tilde{q}}_{\mu }\left(t^{\prime }\right)+\int_{t}^{t^{\prime }}dt^{\prime \prime }\;\frac{{\left(p_{\mu }\right)}_{t^{\prime \prime }}}{m}-Q\;. \end{array}$$
Here, we use the formal solutions (3b) and (3d) (see also (Equation 4)). We moreover note that the solution ([Disp-formula FD3a-entropy-26-00277]) for $p_{\mu }$, i.e., ${\left(p_{\mu }\right)}_{t^{\prime }}=p_{\mu }-\int_{t}^{t^{\prime }}dt^{\prime \prime }\;\frac{\partial H_{\mathrm{bath}}\left(\phi_{t^{\prime \prime }},Q_{t^{\prime \prime }}\right)}{\partial q_{\mu }}$, contains the transformed ${\tilde{q}}_{\mu }\left(t^{\prime }\right)=q_{\mu }-\frac{P}{M}\left(t^{\prime }-t\right)$ in $\frac{\partial H_{\mathrm{int}}\left(q_{t^{\prime \prime }},Q_{t^{\prime \prime }}\right)}{\partial q_{\mu }}$ via its dependence on the differences $q_{\mu }-Q$, and analogously for similar expressions. The transformation to the co-moving frame can therefore be seen as a time-dependent shift $q\to \tilde{q}\left(t^{\prime }\right)$ in the initial *q* value. Since the integral ∫dq in (Equation 27) is over all possible *q* values, and since $dq=d\tilde{q}\left(t^{\prime }\right)$, this is the same for all transformations $\tilde{q}\left(t^{\prime }\right)$, independent of the time point $t^{\prime }$. Within the integral, $\tilde{q}\left(t^{\prime }\right)$ therefore plays the role of a new initial value, i.e., under the integral, we can drop the time argument in $\tilde{q}\left(t^{\prime }\right)$. In the following calculation (second equal sign below), we make this explicit by replacing $\tilde{q}\left(t^{\prime }\right)$ with the symbol *q*, which we used to indicate the initial values at starting time *t* all along. Using the transformation (Equation 28), we apply (29) to rewrite the interaction part (Equation 27) as
(30)$$\begin{array}{cc} & -\frac{1}{\Omega (E)}\int_{t}^{t+\Delta t}dt^{\prime }\;\int d\phi \;\delta \left(E-H_{\mathrm{bath}}\left(\phi ,Q\right)\right)\;\frac{\partial H_{\mathrm{int}}\left(q_{t^{\prime }},Q_{t^{\prime }}\right)}{\partial Q_{i}}\\ & \;=\;-\frac{1}{\Omega (E)}\int_{t}^{t+\Delta t}dt^{\prime }\;\int dp\;d\tilde{q}\left(t^{\prime }\right)\\ & \;\;\;\;\;\times \delta \left(E-H_{\mathrm{bulk}}\left(p,\tilde{q}\left(t^{\prime }\right)\right)-\sum_{\mu }U_{\mathrm{int}}\left(|{\tilde{q}}_{\mu }\left(t^{\prime }\right)-\left[Q-\frac{P}{M}\left(t^{\prime }-t\right)\right]|\right)\right)\;\\ & \;\;\;\;\;\;\;\times \sum_{\mu }\frac{\partial U_{\mathrm{int}}\left(\right|{\tilde{q}}_{\mu }\left(t^{\prime }\right)+\int_{t}^{t^{\prime }}dt^{\prime \prime }\;\frac{{\left(p_{\mu }\right)}_{t^{\prime \prime }}}{m}-Q\left|\right)}{\partial Q_{i}}\\ & \;=\;-\frac{1}{\Omega (E)}\int_{t}^{t+\Delta t}dt^{\prime }\;\int dp\;dq\;\delta \left(E-H_{\mathrm{bulk}}\left(p,q\right)-\sum_{\mu }U_{\mathrm{int}}\left(|q_{\mu }-\left[Q-\frac{P}{M}\left(t^{\prime }-t\right)\right]|\right)\right)\;\\ & \;\;\;\;\;\;\;\times \sum_{\mu }\frac{\partial U_{\mathrm{int}}\left(\right|q_{\mu }+\int_{t}^{t^{\prime }}dt^{\prime \prime }\;\frac{{\left(p_{\mu }\right)}_{t^{\prime \prime }}}{m}-Q\left|\right)}{\partial Q_{i}}\\ & \;=\;-\frac{1}{\Omega (E)}\int_{t}^{t+\Delta t}dt^{\prime }\;\int d\phi \;\delta \left(E-H_{\mathrm{bulk}}\left(\phi \right)-H_{\mathrm{int}}\left(q,Q-\frac{P}{M}\left[t^{\prime }-t\right]\right)\right)\;\frac{\partial H_{\mathrm{int}}\left(q_{t^{\prime }},Q\right)}{\partial Q_{i}}\;. \end{array}$$
The main effect of the transformation to the co-moving frame is thus to move the time dependence in $Q_{t^{\prime }}$ from the interaction force outside the δ function to the interaction Hamiltonian inside the δ function.

In order to extract the leading order in Δt, we introduce $1=-\frac{\partial }{\partial t^{\prime }}\left[\Delta t-\left(t^{\prime }-t\right)\right]$ in the time integral and perform a partial integration (using the “more natural” insertion $1=\frac{\partial }{\partial t^{\prime }}\left(t^{\prime }-t\right)$ rather than $1=-\frac{\partial }{\partial t^{\prime }}\left[\Delta t-\left(t^{\prime }-t\right)\right]$ would lead to non-vanishing boundary terms after partial integration):(31)$$\begin{array}{cc} & -\frac{1}{\Omega (E)}\int_{t}^{t+\Delta t}dt^{\prime }\;\int d\phi \;\delta \left(E-H_{\mathrm{bulk}}\left(\phi \right)-H_{\mathrm{int}}\left(q,Q-\frac{P}{M}\left(t^{\prime }-t\right)\right)\right)\;\frac{\partial H_{\mathrm{int}}\left(q_{t^{\prime }},Q\right)}{\partial Q_{i}}\\ & \;=\;-\frac{1}{\Omega (E)}\int_{t}^{t+\Delta t}dt^{\prime }\;\left[-\frac{\partial }{\partial t^{\prime }}\left[\Delta t-\left(t^{\prime }-t\right)\right]\right]\\ & \;\;\;\;\;\times \int d\phi \;\delta \left(E-H_{\mathrm{bulk}}\left(\phi \right)-H_{\mathrm{int}}\left(q,Q-\frac{P}{M}\left(t^{\prime }-t\right)\right)\right)\;\frac{\partial H_{\mathrm{int}}\left(q_{t^{\prime }},Q\right)}{\partial Q_{i}}\\ & \;=\;\frac{1}{\Omega (E)}{\left[\left[\Delta t-\left(t^{\prime }-t\right)\right]\int d\phi \;\delta \left(E-H_{\mathrm{bulk}}\left(\phi \right)-H_{\mathrm{int}}\left(q,Q-\frac{P}{M}\left(t^{\prime }-t\right)\right)\right)\;\frac{\partial H_{\mathrm{int}}\left(q_{t^{\prime }},Q\right)}{\partial Q_{i}}\right]}_{t^{\prime }=t}^{t+\Delta t}\\ & \;\;\;-\frac{1}{\Omega (E)}\int_{t}^{t+\Delta t}dt^{\prime }\;\left[\Delta t-\left(t^{\prime }-t\right)\right]\int d\phi \;\delta^{\prime }\left(E-H_{\mathrm{bulk}}\left(\phi \right)-H_{\mathrm{int}}\left(q,Q-\frac{P}{M}\left(t^{\prime }-t\right)\right)\right)\;\\ & \;\;\;\;\;\;\;\times \sum_{j}\left[-\frac{\partial H_{\mathrm{int}}\left(q,Q-\frac{P}{M}\left(t^{\prime }-t\right)\right))}{\partial Q_{j}}\left(-\frac{P_{j}}{M}\right)\right]\frac{\partial H_{\mathrm{int}}\left(q_{t^{\prime }},Q\right)}{\partial Q_{i}}\\ & \;\;\;-\frac{1}{\Omega (E)}\int_{t}^{t+\Delta t}dt^{\prime }\;\left[\Delta t-\left(t^{\prime }-t\right)\right]\int d\phi \;\delta \left(E-H_{\mathrm{bulk}}\left(\phi \right)-H_{\mathrm{int}}\left(q,Q-\frac{P}{M}\left(t^{\prime }-t\right)\right)\right)\;\\ & \;\;\;\;\;\;\;\times \sum_{\mu ,j}\frac{\partial^{2}H_{\mathrm{int}}\left(q_{t^{\prime }},Q\right)}{\partial q_{\mu ,j}\partial Q_{i}}{\left({\dot{q}}_{\mu ,j}\right)}_{t^{\prime }}\;. \end{array}$$
The first line in the last equality is the boundary term from the partial integration. In addition, the partial integration produces two new terms (second and third summand) due to the (slow) time dependence in the interaction Hamiltonian within the δ function and the (fast) one in the interaction force outside the δ function.

Next, we transform back to the original coordinates in the laboratory-fixed frame (and use $q_{t}=q$, $Q_{t}=Q$ again):(32)$$\begin{array}{cc} & -\frac{1}{\Omega (E)}\int_{t}^{t+\Delta t}dt^{\prime }\;\int d\phi \;\delta \left(E-H_{\mathrm{bulk}}\left(\phi \right)-H_{\mathrm{int}}\left(q,Q-\frac{P}{M}\left(t^{\prime }-t\right)\right)\right)\;\frac{\partial H_{\mathrm{int}}\left(q_{t^{\prime }},Q\right)}{\partial Q_{i}}\\ & \;=\;-\frac{\Delta t}{\Omega (E)}\int d\phi \;\delta \left(E-H_{\mathrm{bulk}}\left(\phi \right)-H_{\mathrm{int}}\left(q,Q\right)\right)\;\frac{\partial H_{\mathrm{int}}\left(q,Q\right)}{\partial Q_{i}}\\ & \;\;\;-\sum_{j}\frac{P_{j}}{M}\;\frac{1}{\Omega (E)}\int_{t}^{t+\Delta t}dt^{\prime }\;\left[\Delta t-\left(t^{\prime }-t\right)\right]\\ & \;\;\;\;\;\;\;\times \int d\phi \;\delta^{\prime }\left(E-H_{\mathrm{bath}}\left(\phi ,Q\right)\right)\;\frac{\partial H_{\mathrm{int}}\left(q,Q\right)}{\partial Q_{j}}\frac{\partial H_{\mathrm{int}}\left(q_{t^{\prime }},Q_{t^{\prime }}\right)}{\partial Q_{i}}\\ & \;\;\;-\frac{1}{\Omega (E)}\int_{t}^{t+\Delta t}dt^{\prime }\;\left[\Delta t-\left(t^{\prime }-t\right)\right]\int d\phi \;\delta \left(E-H_{\mathrm{bath}}\left(\phi ,Q\right)\right)\;\sum_{\mu ,j}\frac{\partial^{2}H_{\mathrm{int}}\left(q_{t^{\prime }},Q_{t^{\prime }}\right)}{\partial q_{\mu ,j}\partial Q_{i}}\frac{{\left(p_{\mu ,j}\right)}_{t^{\prime }}}{m}\;. \end{array}$$
The first line corresponds to the zeroth-order contribution discussed above (see Equations (Equation 25) and (Equation 26)) and is proven in Appendix A to vanish for homogeneous, translational-invariant environments (in other cases, this term would result in a net force from a so-called ”potential of mean force” [9]). Likewise, we show in Appendix A that phase-space averages of the type like the one in the last line vanish in zeroth order Δt, for which $Q_{t^{\prime }}\approx Q_{t}=Q$ (see Equation (A7)), implying that they contribute only in $O(\Delta t^{2})$. Hence, the only contribution in linear order Δt comes from the term in the second and third line when approximating $Q_{t^{\prime }}\approx Q$. The prime at the δ function denotes the derivative with respect to its argument, or, equivalently, with respect to *E*. Using the definition of temperature (Equation 9), we prove in Appendix B that, for “non-extensive” averages (like averages over short-ranged interactions between the system particle and bath), we are allowed to replace $\delta^{\prime }\left(\ldots \right)$ with $\frac{1}{k_{B}T}\delta \left(\ldots \right)$. We finally find that
(33)$$\begin{array}{cc} & -\frac{1}{\Omega (E)}\int_{t}^{t+\Delta t}dt^{\prime }\;\int d\phi \;\delta \left(E-H_{\mathrm{bath}}\left(\phi ,Q\right)\right)\;\frac{\partial H_{\mathrm{int}}\left(q_{t^{\prime }},Q_{t^{\prime }}\right)}{\partial Q_{i}}\\ & \;=\;-\sum_{j}\frac{P_{j}}{M}\;\frac{1}{k_{B}T}\frac{1}{\Omega (E)}\int_{t}^{t+\Delta t}dt\;\left[\Delta t-\left(t^{\prime }-t\right)\right]\\ & \;\;\;\;\;\;\times \int d\phi \;\delta \left(E-H_{\mathrm{bath}}\left(\phi ,Q\right)\right)\;\frac{\partial H_{\mathrm{int}}\left(q,Q\right)}{\partial Q_{j}}\frac{\partial H_{\mathrm{int}}\left(q_{t^{\prime }},Q\right)}{\partial Q_{i}}+O\left(\Delta t^{2}\right)\\ & \;=\;-\sum_{j}\frac{P_{j}}{M}\Delta t\;\frac{1}{k_{B}T}\int_{t}^{t+\Delta t}dt^{\prime }\;\left(1-\frac{t^{\prime }-t}{\Delta t}\right){\langle \frac{\partial H_{\mathrm{int}}\left(q,Q\right)}{\partial Q_{j}}\frac{\partial H_{\mathrm{int}}\left(q_{t^{\prime }},Q\right)}{\partial Q_{i}}\rangle }_{\;\;\mathrm{mc}}+O\left(\Delta t^{2}\right)\;. \end{array}$$

##### Laboratory Frame

We can, of course, perform exactly the same partial time integration as in the calculation above without transforming to the co-moving frame. Using again the results (A7) from Appendix A, which imply that certain terms do not contribute in linear order Δt (in the third and fourth step below), the main calculation steps are
(34)$$\begin{array}{cc} & -\frac{1}{\Omega (E)}\int_{t}^{t+\Delta t}dt^{\prime }\;\int d\phi \;\delta \left(E-H_{\mathrm{bath}}\left(\phi ,Q\right)\right)\;\frac{\partial H_{\mathrm{int}}\left(q_{t^{\prime }},Q_{t^{\prime }}\right)}{\partial Q_{i}}\\ & \;=\;-\frac{1}{\Omega (E)}\int_{t}^{t+\Delta t}dt^{\prime }\;\left[-\frac{\partial }{\partial t^{\prime }}\left[\Delta t-\left(t^{\prime }-t\right)\right]\right]\int d\phi \;\delta \left(E-H_{\mathrm{bath}}\left(\phi ,Q\right)\right)\;\frac{\partial H_{\mathrm{int}}\left(q_{t^{\prime }},Q_{t^{\prime }}\right)}{\partial Q_{i}}\\ & \;=\;\frac{1}{\Omega (E)}{\left[\left[\Delta t-\left(t^{\prime }-t\right)\right]\int d\phi \;\delta \left(E-H_{\mathrm{bath}}\left(\phi ,Q\right)\right)\;\frac{\partial H_{\mathrm{int}}\left(q_{t^{\prime }},Q_{t^{\prime }}\right)}{\partial Q_{i}}\right]}_{t^{\prime }=t}^{t+\Delta t}\\ & \;\;\;-\frac{1}{\Omega (E)}\int_{t}^{t+\Delta t}dt^{\prime }\;\left[\Delta t-\left(t^{\prime }-t\right)\right]\int d\phi \;\delta \left(E-H_{\mathrm{bath}}\left(\phi ,Q\right)\right)\;\frac{\partial }{\partial t^{\prime }}\frac{\partial H_{\mathrm{int}}\left(q_{t^{\prime }},Q_{{\;}^{\prime }t}\right)}{\partial Q_{i}}\\ & \;=\;O\left(\Delta t^{2}\right)-\frac{1}{\Omega (E)}\int_{t}^{t+\Delta t}dt^{\prime }\;\left[\Delta t-\left(t^{\prime }-t\right)\right]\int d\phi \;\delta \left(E-H_{\mathrm{bath}}\left(\phi ,Q\right)\right)\;\\ & \;\;\;\;\;\;\;\times \left[\sum_{j}\frac{\partial^{2}H_{\mathrm{int}}\left(q_{t^{\prime }},Q_{t^{\prime }}\right)}{\partial Q_{j}\partial Q_{i}}\frac{P_{j}}{M}+\sum_{\mu ,j}\frac{\partial^{2}H_{\mathrm{int}}\left(q_{t^{\prime }},Q_{t^{\prime }}\right)}{\partial q_{\mu ,j}\partial Q_{i}}\frac{{\left(p_{\mu ,j}\right)}_{t^{\prime }}}{m}\right]\\ & \;=\;-\sum_{j}\frac{P_{j}}{M}\Delta t\;\int_{t}^{t+\Delta t}dt^{\prime }\;\left(1-\frac{t^{\prime }-t}{\Delta t}\right)\int d\phi \;\delta \left(E-H_{\mathrm{bath}}\left(\phi ,Q\right)\right)\;\frac{\partial^{2}H_{\mathrm{int}}\left(q_{t^{\prime }},Q_{t^{\prime }}\right)}{\partial Q_{j}\partial Q_{i}}+O\left(\Delta t^{2}\right)\;. \end{array}$$
Finally,
(35)$$\begin{array}{cc} & -\frac{1}{\Omega (E)}\int_{t}^{t+\Delta t}dt^{\prime }\;\int d\phi \;\delta \left(E-H_{\mathrm{bath}}\left(\phi ,Q\right)\right)\;\frac{\partial H_{\mathrm{int}}\left(q_{t^{\prime }},Q_{t^{\prime }}\right)}{\partial Q_{i}}\\ & \;\;\;=\;-\sum_{j}\frac{P_{j}}{M}\Delta t\;\int_{t}^{t+\Delta t}dt^{\prime }\;\left(1-\frac{t^{\prime }-t}{\Delta t}\right){\langle \frac{\partial^{2}H_{\mathrm{int}}\left(q_{t^{\prime }},Q\right)}{\partial Q_{j}\partial Q_{i}}\rangle }_{\;\;\mathrm{mc}}+O\left(\Delta t^{2}\right)\;. \end{array}$$

##### The O(Δt) Contribution in $\langle \Delta P_{i}\rangle$

Summarizing what we found for $\langle \Delta P_{i}\rangle$, the contribution to linear order in Δt is
(36)$$\langle \Delta P_{i}\rangle =f_{i}\left(Q,\lambda \right)\Delta t-\sum_{j}\gamma_{ij}\frac{P_{j}}{M}\Delta t\;.$$
Here, we introduce the friction coefficient $\gamma_{ij}$, given by the two alternative expressions from (Equation 33) and (Equation 35):(37)$$\begin{array}{cc} \gamma_{ij} & =\frac{1}{k_{B}T}\int_{t}^{t+\Delta t}dt^{\prime }\;\left(1-\frac{t^{\prime }-t}{\Delta t}\right){\langle \frac{\partial H_{\mathrm{int}}\left(q,Q\right)}{\partial Q_{j}}\frac{\partial H_{\mathrm{int}}\left(q_{t^{\prime }},Q\right)}{\partial Q_{i}}\rangle }_{\;\;\mathrm{mc}}\\ & =\int_{t}^{t+\Delta t}dt^{\prime }\;\left(1-\frac{t^{\prime }-t}{\Delta t}\right){\langle \frac{\partial^{2}H_{\mathrm{int}}\left(q_{t^{\prime }},Q\right)}{\partial Q_{j}\partial Q_{i}}\rangle }_{\;\;\mathrm{mc}}\;, \end{array}$$
We remark that the equivalence of $\frac{1}{k_{B}T}{\langle \frac{\partial H_{\mathrm{int}}\left(q,Q\right)}{\partial Q_{j}}\frac{\partial H_{\mathrm{int}}\left(q_{t},Q\right)}{\partial Q_{i}}\rangle }_{\;\mathrm{mc}}$ and ${\langle \frac{\partial^{2}H_{\mathrm{int}}\left(q_{t},Q\right)}{\partial Q_{j}\partial Q_{i}}\rangle }_{\;\mathrm{mc}}$ can be verified directly; see Appendix C. It is obvious from the second line that $\gamma_{ij}$ is symmetric under exchange of the indices *i*,*j*, a property that is then also valid for the auto-correlation in the first line.

The first expression represents a time integral over the correlation of the interaction forces, averaged over the various dynamical evolutions of the bath particles in the microcanonical ensemble. By shifting the integration variable $t^{\prime }$ to the new variable $\tilde{t}=t^{\prime }-t$ and by defining ${\tilde{q}}_{\tilde{t}}=q_{\tilde{t}+t}$ (such that ${\tilde{q}}_{0}=q_{t}=q$), the time integration runs over the interval [0,Δt],
(38)$$\begin{array}{cc} \gamma_{ij} & =\frac{1}{k_{B}T}\int_{0}^{\Delta t}dt\;\left(1-\frac{t}{\Delta t}\right){\langle \frac{\partial H_{\mathrm{int}}\left(q,Q\right)}{\partial Q_{j}}\frac{\partial H_{\mathrm{int}}\left(q_{t},Q\right)}{\partial Q_{i}}\rangle }_{\;\;\mathrm{mc}}\;, \end{array}$$
where we drop the tilde symbol to simplify notation. Since we assume that perturbations in the bath decay on the characteristic time scale τ, we expect the correlations to also decay on this time scale, such that ${\langle \frac{\partial H_{\mathrm{int}}\left(q,Q\right)}{\partial Q_{j}}\frac{\partial H_{\mathrm{int}}\left(q_{t},Q\right)}{\partial Q_{i}}\rangle }_{\;\mathrm{mc}}$ vanishes for times t≫τ. As a consequence, we can extend the upper integration limit Δt≫τ to infinity. Moreover, the term t/Δt in the integral is effectively O(τ/Δt) and thus negligibly small, as can be verified by rescaling time according to $\tilde{t}=t/\tau$ (to make $\tilde{t}∼O\left(1\right)$ when t∼O(τ)). Hence, our expression for the friction tensor becomes
(39)$$\gamma_{ij}=\frac{1}{k_{B}T}\int_{0}^{\infty }dt\;{\langle \frac{\partial H_{\mathrm{int}}\left(q,Q\right)}{\partial Q_{j}}\frac{\partial H_{\mathrm{int}}\left(q_{t},Q\right)}{\partial Q_{i}}\rangle }_{\;\;\mathrm{mc}}\;.$$
This is exactly the same result as obtained from projection operator techniques; see, e.g., Chapter 11 in [2].

#### 3.2.3. Evaluation of $\langle \Delta Q_{i}\Delta Q_{j}\rangle$

The evaluation of $\langle \Delta Q_{i}\Delta Q_{j}\rangle$ proceeds along similar lines as the one of $\langle \Delta Q_{i}\rangle$. We use (Equation 10) in (18c) to write
(40)$$\begin{array}{cc} \langle \Delta Q_{i}\Delta Q_{j}\rangle & =\int d\Phi^{\prime }\;\left(Q_{i}^{\prime }-Q_{i}\right)\left(Q_{j}^{\prime }-Q_{j}\right)\;\frac{\int d\phi \;\delta \left(E-H_{\mathrm{bath}}\left(\phi ,Q\right)\right)\;\delta \left(\Phi^{\prime }-\Phi_{\Delta t}\left(\phi ,\Phi \right)\right)}{\Omega (E)}\\ & =\frac{1}{\Omega (E)}\int d\phi \;\left[{\left(Q_{i}\right)}_{\Delta t}\left(\phi ,\Phi \right)-Q_{i}\right]\left[{\left(Q_{j}\right)}_{\Delta t}\left(\phi ,\Phi \right)-Q_{j}\right]\;\delta \left(E-H_{\mathrm{bath}}\left(\phi ,Q\right)\right)\;. \end{array}$$
For ${\left(Q_{i}\right)}_{\Delta t}\left(\phi ,\Phi \right)$, we can again use the expansion ${\left(Q_{i}\right)}_{\Delta t}=Q_{i}+\frac{P_{i}}{M}\Delta t+O\left(\Delta t^{2}\right)$ (see (Equation 4)) so that we obtain
(41)$$\begin{array}{cc} \langle \Delta Q_{i}\Delta Q_{j}\rangle & =\frac{1}{\Omega (E)}\int d\phi \;\left(\frac{P_{i}}{M}\Delta t\right)\;\left(\frac{P_{j}}{M}\Delta t\right)\;\delta \left(E-H_{\mathrm{bath}}\left(\phi ,Q\right)\right)+O\left(\Delta t^{2}\right)\\ & =O(\Delta t^{2})\;. \end{array}$$
As expected, $\langle \Delta Q_{i}\Delta Q_{j}\rangle$ does not contribute in linear order in Δt.

#### 3.2.4. Evaluation of $\langle \Delta P_{i}\Delta P_{j}\rangle$

In order to evaluate $\langle \Delta P_{i}\Delta P_{j}\rangle$, we proceed analogously. We start from (18c) and plug in the transition probability (Equation 10) and then the formal solution (3c):(42)$$\begin{array}{cc} \langle \Delta P_{i}\Delta P_{j}\rangle & =\int d\Phi^{\prime }\;\left(P_{i}^{\prime }-P_{i}\right)\left(P_{j}^{\prime }-P_{j}\right)\;\frac{\int d\phi \;\delta \left(E-H_{\mathrm{bath}}\left(\phi ,Q\right)\right)\;\delta \left(\Phi^{\prime }-\Phi_{\Delta t}\left(\phi ,\Phi \right)\right)}{\Omega (E)}\\ & =\frac{1}{\Omega (E)}\int d\phi \;\left[{\left(P_{i}\right)}_{\Delta t}\left(\phi ,\Phi \right)-P_{i}\right]\left[{\left(P_{j}\right)}_{\Delta t}\left(\phi ,\Phi \right)-P_{j}\right]\;\delta \left(E-H_{\mathrm{bath}}\left(\phi ,Q\right)\right)\\ & =\frac{1}{\Omega (E)}\int d\phi \;\delta \left(E-H_{\mathrm{bath}}\left(\phi ,Q\right)\right)\left[-\int_{t}^{t+\Delta t}dt^{\prime }\;\left(\frac{\partial H_{\mathrm{int}}\left(q_{t^{\prime }},Q_{t^{\prime }}\right)}{\partial Q_{i}}+\frac{\partial H_{\mathrm{sys}}\left(\Phi_{t^{\prime }},\lambda_{t^{\prime }}\right)}{\partial Q_{i}}\right)\right]\\ & \;\;\;\;\;\;\times \left[-\int_{t}^{t+\Delta t}dt^{\prime }\;\left(\frac{\partial H_{\mathrm{int}}\left(q_{t^{\prime }},Q_{t^{\prime }}\right)}{\partial Q_{j}}+\frac{\partial H_{\mathrm{sys}}\left(\Phi_{t^{\prime }},\lambda_{t^{\prime }}\right)}{\partial Q_{j}}\right)\right] \end{array}$$
The terms involving $H_{\mathrm{sys}}\left(\Phi_{t^{\prime }},\lambda_{t^{\prime }}\right)$ represent the external forces (Equation 23) exerted on the system. Recalling that these vary slowly in time on Δt intervals, we obtain
(43)$$\begin{array}{cc} \langle \Delta P_{i}\Delta P_{j}\rangle & =-\frac{1}{\Omega (E)}f_{i}\left(Q,\lambda \right)\Delta t\int_{t}^{t+\Delta t}dt^{\prime }\int d\phi \;\delta \left(E-H_{\mathrm{bath}}\left(\phi ,Q\right)\right)\;\frac{\partial H_{\mathrm{int}}\left(q_{t^{\prime }},Q\right)}{\partial Q_{j}}\\ & \;\;-\frac{1}{\Omega (E)}f_{j}\left(Q,\lambda \right)\Delta t\int_{t}^{t+\Delta t}dt^{\prime }\int d\phi \;\delta \left(E-H_{\mathrm{bath}}\left(\phi ,Q\right)\right)\;\frac{\partial H_{\mathrm{int}}\left(q_{t^{\prime }},Q\right)}{\partial Q_{i}}\\ & \;\;+\frac{1}{\Omega (E)}\int d\phi \;\delta \left(E-H_{\mathrm{bath}}\left(\phi ,Q\right)\right)\;\\ & \;\;\;\;\;\times \left[\int_{t}^{t+\Delta t}dt^{\prime }\;\frac{\partial H_{\mathrm{int}}\left(q_{t^{\prime }},Q\right)}{\partial Q_{i}}\right]\left[\int_{t}^{t+\Delta t}dt^{\prime }\;\frac{\partial H_{\mathrm{int}}\left(q_{t^{\prime }},Q\right)}{\partial Q_{j}}\right]\\ & \;\;+O(\Delta t^{2}) \end{array}$$
The microcanonical averages appearing in the first two lines are the average net forces of the bath on an immobile system particle, as in (Equation 26), and are proven to vanish in Appendix A, cf. Equation ([Disp-formula FD70a-entropy-26-00277]). We thus obtain
(44)$$\langle \Delta P_{i}\Delta P_{j}\rangle ={\langle \int_{t}^{t+\Delta t}dt^{\prime }\int_{t}^{t+\Delta t}dt^{\prime \prime }\;\frac{\partial H_{\mathrm{int}}\left(q_{t^{\prime }},Q\right)}{\partial Q_{i}}\frac{\partial H_{\mathrm{int}}\left(q_{t^{\prime \prime }},Q\right)}{\partial Q_{j}}\rangle }_{\;\;\mathrm{mc}}+O\left(\Delta t^{2}\right)\;,$$

In order to find out if and how this double time integral is related to the friction tensor (Equation 37), our aim is to rewrite it as a single time integral. We first split up the square integration domain into two triangular domains and then change the order of integration in one of the domains:(45)$$\begin{array}{cc} & {\langle \int_{t}^{t+\Delta t}dt^{\prime }\int_{t}^{t+\Delta t}dt^{\prime \prime }\;\frac{\partial H_{\mathrm{int}}\left(q_{t^{\prime }},Q\right)}{\partial Q_{i}}\frac{\partial H_{\mathrm{int}}\left(q_{t^{\prime \prime }},Q\right)}{\partial Q_{j}}\rangle }_{\;\;\mathrm{mc}}\\ & \;\;=\int_{t}^{t+\Delta t}dt^{\prime }\int_{t}^{t^{\prime }}dt^{\prime \prime }\;{\langle \frac{\partial H_{\mathrm{int}}\left(q_{t^{\prime }},Q\right)}{\partial Q_{i}}\frac{\partial H_{\mathrm{int}}\left(q_{t^{\prime \prime }},Q\right)}{\partial Q_{j}}\rangle }_{\;\;\mathrm{mc}}\\ & \;\;\;\;\;+\int_{t}^{t+\Delta t}dt^{\prime }\int_{t^{\prime }}^{t+\Delta t}dt^{\prime \prime }\;{\langle \frac{\partial H_{\mathrm{int}}\left(q_{t^{\prime }},Q\right)}{\partial Q_{i}}\frac{\partial H_{\mathrm{int}}\left(q_{t^{\prime \prime }},Q\right)}{\partial Q_{j}}\rangle }_{\;\;\mathrm{mc}}\\ & \;\;=\int_{t}^{t+\Delta t}dt^{\prime \prime }\int_{t^{\prime \prime }}^{t+\Delta t}dt^{\prime }\;{\langle \frac{\partial H_{\mathrm{int}}\left(q_{t^{\prime }},Q\right)}{\partial Q_{i}}\frac{\partial H_{\mathrm{int}}\left(q_{t^{\prime \prime }},Q\right)}{\partial Q_{j}}\rangle }_{\;\;\mathrm{mc}}\\ & \;\;\;\;\;+\int_{t}^{t+\Delta t}dt^{\prime }\int_{t^{\prime }}^{t+\Delta t}dt^{\prime \prime }\;{\langle \frac{\partial H_{\mathrm{int}}\left(q_{t^{\prime }},Q\right)}{\partial Q_{i}}\frac{\partial H_{\mathrm{int}}\left(q_{t^{\prime \prime }},Q\right)}{\partial Q_{j}}\rangle }_{\;\;\mathrm{mc}}\\ & \;\;=\int_{t}^{t+\Delta t}dt^{\prime }\int_{t^{\prime }}^{t+\Delta t}dt^{\prime \prime }\;[{\langle \frac{\partial H_{\mathrm{int}}\left(q_{t^{\prime \prime }},Q\right)}{\partial Q_{i}}\frac{\partial H_{\mathrm{int}}\left(q_{t^{\prime }},Q\right)}{\partial Q_{j}}\rangle }_{\;\;\mathrm{mc}}\\ & \;\;\;\;\;\;\;\;+{\langle \frac{\partial H_{\mathrm{int}}\left(q_{t^{\prime }},Q\right)}{\partial Q_{i}}\frac{\partial H_{\mathrm{int}}\left(q_{t^{\prime \prime }},Q\right)}{\partial Q_{j}}\rangle }_{\;\;\mathrm{mc}}\;]\;. \end{array}$$
The last equality is obtained by renaming the integration variables $t^{\prime }↔t^{\prime \prime }$ in the first term. In the next step, we shift the integration interval in the inner integral by introducing the new variable $\tilde{t}=t^{\prime \prime }-\left(t^{\prime }-t\right)$:(46)$$\begin{array}{cc} & {\langle \int_{t}^{t+\Delta t}dt^{\prime }\int_{t}^{t+\Delta t}dt^{\prime \prime }\;\frac{\partial H_{\mathrm{int}}\left(q_{t^{\prime }},Q\right)}{\partial Q_{i}}\frac{\partial H_{\mathrm{int}}\left(q_{t^{\prime \prime }},Q\right)}{\partial Q_{j}}\rangle }_{\;\;\mathrm{mc}}\\ & \;\;=\int_{t}^{t+\Delta t}dt^{\prime }\int_{t}^{t+\Delta t-(t^{\prime }-t)}d\tilde{t}\;[{\langle \frac{\partial H_{\mathrm{int}}\left(q_{t^{\prime }+\left(\tilde{t}-t\right)},Q\right)}{\partial Q_{i}}\frac{\partial H_{\mathrm{int}}\left(q_{t^{\prime }},Q\right)}{\partial Q_{j}}\rangle }_{\;\;\mathrm{mc}}\\ & \;\;\;\;\;\;\;\;\;+{\langle \frac{\partial H_{\mathrm{int}}\left(q_{t^{\prime }},Q\right)}{\partial Q_{i}}\frac{\partial H_{\mathrm{int}}\left(q_{t^{\prime }+\left(\tilde{t}-t\right)},Q\right)}{\partial Q_{j}}\rangle }_{\;\;\mathrm{mc}}\;]\;. \end{array}$$
The correlations inside the integrals are now expressed in terms of the bath states at time $t^{\prime }$ and at the later time $t^{\prime }+\left(\tilde{t}-t\right)$. Due to our central assumption that the bath is in equilibrium, these correlations are functions of the time difference only, i.e., they depend on $\tilde{t}-t$ but not $t^{\prime }$. We are thus free to choose any time point for $t^{\prime }$, and we fix it to the initial time *t* of the considered time interval Δt (remember that the initial bath state is $q_{t}=q$):(47)$$\begin{array}{cc} & {\langle \int_{t}^{t+\Delta t}dt^{\prime }\int_{t}^{t+\Delta t}dt^{\prime \prime }\;\frac{\partial H_{\mathrm{int}}\left(q_{t^{\prime }},Q\right)}{\partial Q_{i}}\frac{\partial H_{\mathrm{int}}\left(q_{t^{\prime \prime }},Q\right)}{\partial Q_{j}}\rangle }_{\;\;\mathrm{mc}}\\ & \;\;=\int_{t}^{t+\Delta t}dt^{\prime }\int_{t}^{t+\Delta t-(t^{\prime }-t)}d\tilde{t}\;[{\langle \frac{\partial H_{\mathrm{int}}\left(q_{\tilde{t}},Q\right)}{\partial Q_{i}}\frac{\partial H_{\mathrm{int}}\left(q,Q\right)}{\partial Q_{j}}\rangle }_{\;\;\mathrm{mc}}\\ & \;\;\;\;\;\;\;\;\;+{\langle \frac{\partial H_{\mathrm{int}}\left(q,Q\right)}{\partial Q_{i}}\frac{\partial H_{\mathrm{int}}\left(q_{\tilde{t}},Q\right)}{\partial Q_{j}}\rangle }_{\;\;\mathrm{mc}}\;]\;. \end{array}$$
Exploiting the i↔j symmetry of the correlations (cf. Equation (Equation 37)), the two terms can now be combined into one. Moreover, since the correlations no longer depend on $t^{\prime }$, we can perform the $t^{\prime }$ integral by exchanging the order of integration:(48)$$\begin{array}{cc} & {\langle \int_{t}^{t+\Delta t}dt^{\prime }\int_{t}^{t+\Delta t}dt^{\prime \prime }\;\frac{\partial H_{\mathrm{int}}\left(q_{t^{\prime }},Q\right)}{\partial Q_{i}}\frac{\partial H_{\mathrm{int}}\left(q_{t^{\prime \prime }},Q\right)}{\partial Q_{j}}\rangle }_{\;\;\mathrm{mc}}\\ & \;\;=2\int_{t}^{t+\Delta t}d\tilde{t}\int_{t}^{t+\Delta t-(\tilde{t}-t)}dt^{\prime }\;{\langle \frac{\partial H_{\mathrm{int}}\left(q_{\tilde{t}},Q\right)}{\partial Q_{i}}\frac{\partial H_{\mathrm{int}}\left(q,Q\right)}{\partial Q_{j}}\rangle }_{\;\;\mathrm{mc}}\\ & \;\;=2\int_{t}^{t+\Delta t}d\tilde{t}\;\left[\Delta t-\left(\tilde{t}-t\right)\right]{\langle \frac{\partial H_{\mathrm{int}}\left(q_{\tilde{t}},Q\right)}{\partial Q_{i}}\frac{\partial H_{\mathrm{int}}\left(q,Q\right)}{\partial Q_{j}}\rangle }_{\;\;\mathrm{mc}}\;. \end{array}$$

In linear order in Δt, we therefore find
(49)$$\begin{array}{cc} \langle \Delta P_{i}\Delta P_{j}\rangle & =2\Delta t\int_{t}^{t+\Delta t}dt^{\prime }\;\left(1-\frac{t^{\prime }-t}{\Delta t}\right)\;{\langle \frac{\partial H_{\mathrm{int}}\left(q,Q\right)}{\partial Q_{j}}\frac{\partial H_{\mathrm{int}}\left(q_{t^{\prime }},Q\right)}{\partial Q_{i}}\rangle }_{\;\;\mathrm{mc}}\;. \end{array}$$
Finally, with the definition (Equation 37) of the friction coefficient, we can write this result in the suggestive form
(50)$$\langle \Delta P_{i}\Delta P_{j}\rangle =2k_{B}T\gamma_{ij}\;\Delta t\;.$$

#### 3.2.5. Evaluation of $\langle \Delta Q_{i}\Delta P_{j}\rangle$

Finally, we quickly verify that $\langle \Delta Q_{i}\Delta P_{j}\rangle$ has a quadratic (and higher) contribution in Δt. We start from (18e), plug in the expression (Equation 10) for $p_{E}\left(\Phi^{\prime }|\Phi \right)$ and use the formal solutions (3c), (3d) to assess the various contributions:(51)$$\begin{array}{cc} \langle \Delta Q_{i}\Delta P_{j}\rangle & =\int d\Phi^{\prime }\;\left(Q_{i}^{\prime }-Q_{i}\right)\left(P_{j}^{\prime }-P_{j}\right)\;\frac{\int d\phi \;\delta \left(E-H_{\mathrm{bath}}\left(\phi ,Q\right)\right)\;\delta \left(\Phi^{\prime }-\Phi_{\Delta t}\left(\phi ,\Phi \right)\right)}{\Omega (E)}\\ & =\frac{1}{\Omega (E)}\int d\phi \;\left[{\left(Q_{i}\right)}_{\Delta t}\left(\phi ,\Phi \right)-Q_{i}\right]\left[{\left(P_{j}\right)}_{\Delta t}\left(\phi ,\Phi \right)-P_{j}\right]\;\delta \left(E-H_{\mathrm{bath}}\left(\phi ,Q\right)\right)\\ & =\frac{1}{\Omega (E)}\int d\phi \;\delta \left(E-H_{\mathrm{bath}}\left(\phi ,Q\right)\right)\left[\int_{t}^{t+\Delta t}dt^{\prime }\;\frac{{\left(P_{i}\right)}_{t^{\prime }}}{M}\right]\\ & \;\;\times \left[-\int_{t}^{t+\Delta t}dt^{\prime }\;\left(\frac{\partial H_{\mathrm{int}}\left(q_{t^{\prime }},Q_{t^{\prime }}\right)}{\partial Q_{j}}+\frac{\partial H_{\mathrm{sys}}\left(\Phi_{t^{\prime }},\lambda_{t^{\prime }}\right)}{\partial Q_{j}}\right)\right]\;. \end{array}$$
Expanding the slowly varying system position ${\left(Q_{i}\right)}_{t^{\prime }}$ at lowest order Δt, we can approximate $\int_{t}^{t+\Delta t}dt^{\prime }\frac{{\left(P_{i}\right)}_{t^{\prime }}}{M}$ by $\frac{P_{i}}{M}\Delta t$ (cf. also (Equation 20)). We are then basically left with an average over $\Delta P_{j}$, for which we can re-use the results from Section 3.2.2. We thus obtain
(52)$$\begin{array}{cc} \langle \Delta Q_{i}\Delta P_{j}\rangle & =\frac{1}{\Omega (E)}\int d\phi \;\delta \left(E-H_{\mathrm{bath}}\left(\phi ,Q\right)\right)\left[\frac{P_{i}}{M}\Delta t\right]\\ & \;\;\times [-\int_{t}^{t+\Delta t}dt^{\prime }\;\left[\Delta t-\left(t^{\prime }-t\right)\right]\sum_{k}\frac{\partial^{2}H_{\mathrm{int}}\left(q_{t^{\prime }},Q\right)}{\partial Q_{k}\partial Q_{j}}\frac{P_{k}}{M}\\ & \;\;\;\;\;\;-\frac{\partial H_{\mathrm{int}}\left(q,Q\right)}{\partial Q_{j}}\Delta t-\frac{\partial H_{\mathrm{sys}}\left(\Phi ,\lambda \right)}{\partial Q_{j}}\Delta t]+O\left(\Delta t^{2}\right)\\ & =O(\Delta t^{2})\;. \end{array}$$

### 3.3. Summary

Collecting all contributions that are linear in Δt, we have
(53a)$$\begin{array}{cc} \langle \Delta Q_{i}\rangle & =\frac{P_{i}}{M}\Delta t\;, \end{array}$$
(53b)$$\begin{array}{cc} \langle \Delta P_{i}\rangle & =-\sum_{j}\gamma_{ij}\frac{P_{j}}{M}\Delta t+f_{i}\left(Q,\lambda \right)\Delta t\;, \end{array}$$
(53c)$$\begin{array}{cc} \langle \Delta P_{i}\Delta P_{j}\rangle & =2k_{B}T\;\gamma_{ij}\Delta t\;, \end{array}$$
with the tensorial friction coefficient
(54)$$\gamma_{ij}=\frac{1}{k_{B}T}\int_{0}^{\infty }dt\;{\langle \frac{\partial H_{\mathrm{int}}\left(q,Q\right)}{\partial Q_{j}}\frac{\partial H_{\mathrm{int}}\left(q_{t},Q\right)}{\partial Q_{i}}\rangle }_{\;\;\mathrm{mc}}$$
All other averages or correlations are at least $O(\Delta t^{2})$.

It is straightforward to verify that a standard Langevin equation of the form
(55a)$$\begin{array}{cc} {\dot{Q}}_{i}\left(t\right) & =\frac{P_{i}\left(t\right)}{M}\;, \end{array}$$
(55b)$$\begin{array}{cc} {\dot{P}}_{i}\left(t\right) & =-\sum_{j}\gamma_{ij}\frac{P_{j}\left(t\right)}{M}+f_{i}\left(Q\left(t\right),\lambda \left(t\right)\right)+\sqrt{2k_{B}T}\;\sum_{j}{\left(\gamma^{1/2}\right)}_{\;ij}\xi_{j}\left(t\right)\;, \end{array}$$
with mutually independent, unbiased, δ-correlated white noise processes $\xi_{j}\left(t\right)$, generates identical average “displacements” and correlations in linear order Δt (remember that the averages in (53) are conditioned on the initial configuration at the beginning of the time step Δt). Moreover, the diffusion coefficient quantifying the mean square displacement of the spatial coordinates *Q* is
(56)$$D_{ij}=k_{B}T\;{\left(\gamma^{-1}\right)}_{ij}\;,$$
obeying Einstein’s fluctuation–dissipation relation [2,3,4,5].

Strictly speaking, we did not demonstrate that the fluctuations behind the averages (53) are *Gaussian*. However, with the huge separation of time scales τ≪Δt, we can assume that there exists an “intermediate” time scale $\tilde{\Delta t}$, with $\tau \ll \tilde{\Delta t}\ll \Delta t$. It is still much larger than τ, so all the above results apply to displacements over $\tilde{\Delta t}$ as well, but it is considerably smaller than Δt, so displacements over Δt are the result of many independent and identically distributed displacements over $\tilde{\Delta t}$. According to the central limit theorem, the resulting dynamics on Δt scales are then Gaussian-distributed with the moments and correlations given above in (53).

### 3.4. Rules of Calculus

Since we find that the statistical properties of the system momentum are generated by a white noise source, the question about the proper rules of (stochastic) calculus naturally arises, eminent in the procedure of calculating quantities like $\langle P_{i}\;\Delta P_{j}\rangle$. We approach this question by considering averages of the form $\langle g\left(\Phi^{\prime }\right)-g\left(\Phi \right)\rangle$ in lowest order Δt, with an arbitrary (smooth) function *g* of the system state. In analogy to (Equation 10), the probability of observing a certain value $g(\Phi^{\prime })$ is given as
(57)$$p_{E}\left(g\left(\Phi^{\prime }\right)|\Phi \right)=\frac{\int d\phi \;\delta \left(E-H_{\mathrm{bath}}\left(\phi ,Q\right)\right)\;\delta \left(g\left(\Phi^{\prime }\right)-g\left(\Phi_{\Delta t}\left(\phi ,\Phi \right)\right)\right)}{\Omega (E)}\;,$$
such that we obtain
(58)$$\langle g\left(\Phi^{\prime }\right)-g\left(\Phi \right)\rangle =\frac{1}{\Omega (E)}\int d\phi \;\left[g(\Phi_{\Delta t}\left(\phi ,\Phi \right))-g\left(\Phi \right)\right]\;\delta \left(E-H_{\mathrm{bath}}\left(\phi ,Q\right)\right)\;.$$
Using our insights gained above, in Section 3.2, we expand this expression to lowest order in Δt,
(59)$$\begin{array}{cc} \langle g\left(\Phi^{\prime }\right)-g\left(\Phi \right)\rangle & =\frac{1}{\Omega (E)}\int d\phi \;\left[\frac{\partial g(\Phi )}{\partial Q}\cdot \Delta Q+\frac{\partial g(\Phi )}{\partial P}\cdot \Delta P\right.\\ & \;\;\;\;\left.+\frac{1}{2}\sum_{i,j}\frac{\partial^{2}g\left(\Phi \right)}{\partial P_{i}\partial P_{j}}\Delta P_{i}\Delta P_{j}\right]\;\delta \left(E-H_{\mathrm{bath}}\left(\phi ,Q\right)\right)+O\left(\Delta t^{2}\right)\\ & =\frac{\partial g(\Phi )}{\partial Q}\cdot \langle \Delta Q\rangle +\frac{\partial g(\Phi )}{\partial P}\cdot \langle \Delta P\rangle +\frac{1}{2}\sum_{i,j}\frac{\partial^{2}g\left(\Phi \right)}{\partial P_{i}\partial P_{j}}\langle \Delta P_{i}\Delta P_{j}\rangle +O\left(\Delta t^{2}\right)\\ & =\frac{\partial g(\Phi )}{\partial Q}\cdot \langle \Delta Q\rangle +\frac{\partial g(\Phi )}{\partial P}\cdot \langle \Delta P\rangle +k_{B}T\sum_{i,j}\frac{\partial^{2}g\left(\Phi \right)}{\partial P_{i}\partial P_{j}}\gamma_{ij}\Delta t+O\left(\Delta t^{2}\right)\;, \end{array}$$
where we use (Equation 50) in the last step. With the standard rules of (“non-stochastic”) calculus, the expansion in lowest order would result only in the first two terms. The third term corresponds exactly to the additional contribution characteristic for the so-called Stratonovich rule of stochastic calculus [3,4]. Denoting this rule by ∘, we can therefore write
(60)$$\langle g\left(\Phi^{\prime }\right)-g\left(\Phi \right)\rangle =\langle \Delta g\left(\Phi \right)\rangle =\langle \frac{\partial g(\Phi )}{\partial \Phi }\circ \Delta \Phi \rangle \;.$$
As a concrete example, we consider $g\left(\Phi \right)=P^{2}$:(61)$$\langle \Delta \left(P^{2}\right)\rangle =2\langle P\circ \Delta P\rangle =2P\cdot \langle \Delta P\rangle +2k_{B}T\mathrm{Tr}\left(\gamma \right)\Delta t\;.$$

### 3.5. Connection to the Fokker-Planck Equation

In Section 3.1, we prove that the effective system dynamics are Markovian on Δt time scales; see Equation (Equation 12). Therefore, the dynamics also obey the Chapman–Kolmogorov equation [2,3,5], which, by the standard Kramers–Moyal expansion [2,3,5], is recast into the (forward) Kolmogorov equation (adjusted to our notation here),
(62a)$$\begin{array}{c} \frac{\partial }{\partial t^{\prime }}p\left(\Phi^{\prime },t^{\prime }|\Phi ,t\right)=-\sum_{i}\frac{\partial }{\partial P_{i}^{\prime }}A_{i}\left(\Phi^{\prime },t^{\prime }\right)p\left(\Phi^{\prime },t^{\prime }|\Phi ,t\right)-\sum_{i}\frac{\partial }{\partial Q_{i}^{\prime }}B_{i}\left(\Phi^{\prime },t^{\prime }\right)p\left(\Phi^{\prime },t^{\prime }|\Phi ,t\right)\\ +\frac{1}{2}\sum_{i,j}\frac{\partial^{2}}{\partial P_{i}^{\prime }\partial P_{j}^{\prime }}A_{ij}\left(\Phi^{\prime },t^{\prime }\right)p\left(\Phi^{\prime },t|\Phi ,t\right)+\frac{1}{2}\sum_{i,j}\frac{\partial^{2}}{\partial Q_{i}^{\prime }\partial Q_{j}^{\prime }}B_{ij}\left(\Phi^{\prime },t^{\prime }\right)p\left(\Phi^{\prime },t^{\prime }|\Phi ,t\right)\;, \end{array}$$
with the coefficients (using $t^{\prime }-t=\Delta t$ as before)
(62b)$$\begin{array}{cc} A_{i}\left(\Phi ,t\right) & =\lim_{\Delta t\to 0}\frac{1}{\Delta t}\int d\Phi^{\prime }\;p\left(\Phi^{\prime },t^{\prime }|\Phi ,t\right)\left(P_{i}^{\prime }-P_{i}\right)\;, \end{array}$$
(62c)$$\begin{array}{cc} B_{i}\left(\Phi ,t\right) & =\lim_{\Delta t\to 0}\frac{1}{\Delta t}\int d\Phi^{\prime }\;p\left(\Phi^{\prime },t^{\prime }|\Phi ,t\right)\left(Q_{i}^{\prime }-Q_{i}\right)\;, \end{array}$$
(62d)$$\begin{array}{cc} A_{ij}\left(\Phi ,t\right) & =\lim_{\Delta t\to 0}\frac{1}{\Delta t}\int d\Phi^{\prime }\;p\left(\Phi^{\prime },t^{\prime }|\Phi ,t\right)\left(P_{i}^{\prime }-P_{i}\right)\left(P_{j}^{\prime }-P_{j}\right)\;, \end{array}$$
(62e)$$\begin{array}{cc} B_{ij}\left(\Phi ,t\right) & =\lim_{\Delta t\to 0}\frac{1}{\Delta t}\int d\Phi^{\prime }\;p\left(\Phi^{\prime },t^{\prime }|\Phi ,t\right)\left(Q_{i}^{\prime }-Q_{i}\right)\left(Q_{j}^{\prime }-Q_{j}\right)\;. \end{array}$$
Here, $p(\Phi^{\prime },t^{\prime }|\Phi ,t)$ is the transition probability density from Equation (Equation 10) (including explicitly the time arguments, but skipping the subscript *E*). The coefficients (62) are given by exactly the moments we already calculated, cf. Equation (18). From the results listed in Equation (53) (and by remembering that all other moments are $O(\Delta t^{2})$), we immediately obtain
(63a)$$\begin{array}{cc} A_{i}\left(\Phi ,t\right) & =-\sum_{j}\gamma_{ij}\frac{P_{j}}{M}+f_{i}\left(Q,\lambda \right)\;, \end{array}$$
(63b)$$\begin{array}{cc} B_{i}\left(\Phi ,t\right) & =\frac{P_{i}}{M}\;, \end{array}$$
(63c)$$\begin{array}{cc} A_{ij}\left(\Phi ,t\right) & =2k_{B}T\;\gamma_{ij}\;, \end{array}$$
(63d)$$\begin{array}{cc} B_{ij}\left(\Phi ,t\right) & =0\;. \end{array}$$
The forward Kolmogorov Equation ([Disp-formula FD62a-entropy-26-00277]) thus reduces to the standard Fokker–Planck equation for underdamped Brownian motion [2,3,4,5].

## 4. Discussion and Conclusions

Starting from first principles with a full microscopic description of a huge thermal bath (“environment”), interacting with a small system of interest, we demonstrate that the effective equation of motion for the system degrees of freedom alone is given by the standard Langevin Equation (55). To the best of our knowledge, the approach presented here is new. It is based on two assumptions: (i) the bath is in thermal equilibrium at all times, and (ii) there is a vast separation of time scales between the bath and system, i.e., the system degrees of freedom are slow coordinates that evolve on time scales much larger than the time scales characterizing the bath dynamics (e.g., molecular collision times). These are the standard conditions for which the Langevin equation is known to be valid as an effective description of the system dynamics. Accordingly, they sooner or later enter other known derivations of the Langevin equation, like the famous projection operator technique (see, for instance, [1,2,10,11,12,13,14]), or the intriguing procedure presented in Section 3.4.1 of Reference [15].

Here, we take these assumptions as a starting point and, in that way, are able to assess the statistical properties of the system dynamics without directly projecting or integrating out [16] the fast bath degrees of freedom. Moreover, the microscopic origin of the various contributions in the Langevin equation becomes evident. In particular, the friction coefficient is identified with the correlations of the interaction forces (see Equation (Equation 39)), and the fluctuation–dissipation relation (Equation 56) is traced back to the microcanonical definition of temperature (Equation 9). Moreover, the pertinent rules of stochastic calculus are identified by our approach to follow the Stratonovich prescription (see Section 3.4). Our derivation can be straightforwardly generalized to many Brownian particles interacting via pairwise interaction potentials as long as “hydrodynamic” interactions mediated by the thermal bath are neglected.

It will be interesting to explore how our approach generalizes to other situations (e.g., the presence of magnetic fields), to other configuration spaces (e.g., rotational Brownian motion [2], or, more generally, Brownian motion on manifolds) or to other system–bath complexes that can be described by Hamilton’s equations of motion (e.g., classical spin systems [17]).

## Figures and Tables

**Figure 1 entropy-26-00277-f001:**
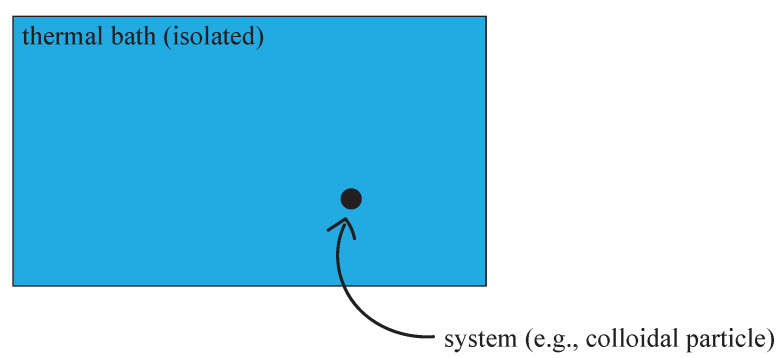
The setup considered here consists of a thermal bath and a “system” in contact with that bath, e.g., a colloidal particle suspended in water at room temperature. The bath is isolated from the rest of the universe and can interact only with the system. The system can be manipulated by external means (e.g., via an externally applied potential), which, however, do not affect the bath directly. (image credit: Jasper Eichhorn).

## Data Availability

The data can be shared up on request.

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
