# Peer review of "Derivation of the Langevin Equation from the Microcanonical Ensemble"

_entropy, 2024, doi:10.3390/e26040277_

Round 1
Reviewer 1 Report
Comments and Suggestions for Authors
The author presents a fine study on deriving the Langevin description from the total Hamiltonian dynamics of the system along with Microcanonical ensemble equilibrium description of the bath.
The paper should be published as is.
An optional recommendation is that the author may want to provide some ideas about how this description would change if there is a constant power injection to the system (say such as Active drive). This essentially drives the system away from just Hamiltonian dynamics, while the bath can still be in equilibrium.
Author Response
The Referee brings up a very interesting point, which I have been thinking about quite a bit. Unfortunately, as of now I do not have a clear idea how to properly modify this description to include a constant power injection to the system of interest. It is trivially possible to add power injection by increasing the total energy (bath + system) with a constant rate. However, as long as this power injection does not couple to the dynamics of the system, there are no observable consequences. To get an active drive, I suspect, one would have to add some kind of "internal degree of freedom" to the system which fuels its translational degrees of freedom. These are very speculative ideas, I am not confident enough about to include them into the manuscript.
Reviewer 2 Report
Comments and Suggestions for Authors
This is clear presentation of a neat derivation of the highly expected result: the validity of the Langevin and Fokker-Planck equations.
I propose to connect to works of J. Roerdink around 1980, who also elimanted fast variables in certain statostical systems.
Author Response
Searching for J. Roerdink's work from the 1980ies I found just one paper on elimination of fast variables:
Roerdink & Weyland, "A generalized Fokker-Planck equation in the case of the Volterra model", Bulletin Math Biol 43, 69 (1981).
Even though this is a very nice and interesting paper, I decided to not include it as a reference, following the advice (VI) from the editors of Entropy:
"If the reviewer(s) recommended references, please critically analyze them to ensure that their inclusion would enhance your manuscript. If you believe these references are unnecessary, you should not include them."
The reasons for that decision are two-fold: (i) The paper considers a model of interacting species from theoretical ecology, rather than the Langevin equation. (ii) It uses projection operator techniques, for which I already cite the articles (books) where this technique is used to derive the Langevin equation.